# Operando X-ray photoelectron spectroscopy of solid electrolyte interphase formation and evolution in $Li_2S$-$P_2S_5$ solid-state electrolytes

Kevin N. Wood [1], K. Xerxes Steirer[2], Simon E. Hafner[3], Chunmei Ban [1], Shriram Santhanagopalan[1], Se-Hee Lee[3] & Glenn Teeter[1]

Solid-state electrolytes such as $Li_2S$-$P_2S_5$ compounds are promising materials that could enable Li metal anodes. However, many solid-state electrolytes are unstable against metallic lithium, and little is known about the chemical evolution of these interfaces during cycling, hindering the rational design of these materials. In this work, operando X-ray photoelectron spectroscopy and real-time in situ Auger electron spectroscopy mapping are developed to probe the formation and evolution of the $Li/Li_2S$-$P_2S_5$ solid-electrolyte interphase during electrochemical cycling, and to measure individual overpotentials associated with specific interphase constituents. Results for the $Li/Li_2S$-$P_2S_5$ system reveal that electrochemically driving $Li^+$ to the surface leads to phase decomposition into $Li_2S$ and $Li_3P$. Additionally, oxygen contamination within the $Li_2S$-$P_2S_5$ leads initially to $Li_3PO_4$ phase segregation, and subsequently to $Li_2O$ formation. The spatially non-uniform distribution of these phases, coupled with differences in their ionic conductivities, have important implications for the overall properties and performance of the solid-electrolyte interphase.

[1] National Renewable Energy Laboratory, 15013 Denver West Pkwy., Golden, CO 80401, USA. [2] Colorado School of Mines, 1500 Illinois St., Golden, CO 80401, USA. [3] University of Colorado, 596 UCB, Boulder, CO 80309, USA. Correspondence and requests for materials should be addressed to K.N.W. (email: kevin.wood@nrel.gov) or to G.T. (email: glenn.teeter@nrel.gov)

As global energy consumption continues to increase rapidly, scalable, safe, and cost-effective strategies for energy storage have become imperative. In pursuit of this goal, numerous 'beyond Li-ion battery' solutions have been intensively researched over the past decade[1, 2]. Many of these next-generation battery architectures use Li metal anodes, which enable dramatically improved theoretical energy densities (gravimetric and volumetric) compared to the current state-of-the-art. A significant challenge is that the extreme reactivity of Li metal tends to cause undesirable side reactions between the Li metal and the electrolyte[3]. In the case of liquid electrolyte systems this often leads to Li consumption, dendrite formation, and the potential for catastrophic failure and fires[4–7].

One widely studied approach for improving the safety of next-generation batteries is the use of solid-state electrolyte (SSE) materials[8]. SSEs improve safety by eliminating flammable liquids, and by providing a physical barrier to dendrite propagation. On the other hand, SSE conductivities are typically lower than liquid electrolytes and nearly all SSE materials are unstable against Li metal. This instability results in degradation of SSE/Li interfaces, creating large interfacial resistances that severely compromise battery performance[9]. Therefore, the interfacial interactions between Li and the SSE must be thoroughly investigated to enable the rational design of stable next-generation battery interfaces. Unfortunately, because of the numerous challenges associated with performing detailed chemical analyses on these types of interfaces[9–12], to date few experimental studies have been reported on their chemical composition and structure.

A current state-of-the-art SSE is sputtered lithium phosphorus oxynitride (LiPON). This material is known for its moderate interfacial stability[9, 13, 14], but relatively poor room-temperature ionic conductivity ($\sim 10^{-6}$ S/cm)[15]. In order to compensate for its low ionic conductivity, much research on LiPON focuses on thin-film fabrication techniques and applications. To circumvent the synthesis challenges of thin film SSEs, researchers have begun to explore higher conductivity systems, like $Li_{10}GeP_2S_{12}$ (LGPS) and $Li_2S–P_2S_5$ (LPS) where conductivities exceeding 5 mS/cm have been demonstrated[16, 17]. Although there is great promise in these sulfur-based SSEs, these highly reactive materials have unstable interfaces with Li, leading to worse rate capabilities than LiPON, even though the initial bulk conductivities are higher[18, 19]. Therefore, the future of sulfur-based SSEs depends on engineering more stable SSE/Li interfaces that enable both high rate capability and extended cycle life.

To optimize these highly conductive SSEs and/or design interfacial barrier materials that enable next-generation SSE battery architectures, a critical first step is understanding how the solid electrolyte interphase (SEI) forms and evolves both chemically and morphologically[3]. Unfortunately, since the SEI is a buried interface (and therefore not readily accessible to the majority of standard analytical techniques) these issues are difficult to elucidate experimentally. A practical challenge that constrains interfacial battery characterization experiments is the extreme reactivity of Li metal, the SSE, and even SEI phases to oxygen, moisture, and organic species. These reactivities limit the utility of typical preparation methods (e.g., focused ion-beam milling, mechanical polishing, etc.) for studying buried interfaces, because such methods can damage, smear, or otherwise fundamentally alter these highly reactive interfaces[20]. Preparing a sample for characterization where the SSE/Li interface is representative of the chemical reactions occurring during operation is extremely challenging and resulting artifacts might limit the utility of such data.

Recently, an in situ X-ray photoelectron spectroscopy (XPS) study by Wenzel et al. helped elucidate the reaction between Li and single-crystal $Li_7P_3S_{11}$ by providing chemical state information before, during, and after physical vapor deposition of Li metal on the surface[11]. These measurements clearly showed that $Li_7P_3S_{11}$ reacts with Li metal to form $Li_2S$ and $Li_3P$ decomposition phases. However, samples in this study were not subjected to electrochemical bias during XPS measurements, and the chemical composition and evolution of the SEI under battery operating conditions was not probed. Therefore, to shed light on how the SEI evolves chemically during cycling, we provide operando measurements of an Li/LPS interface. In this experiment, an electron gun bias drives $Li^+$ migration, while XPS measurements monitor changes at the exposed surface. These data provide detailed compositional and chemical-state information related to SEI formation at the Li/LPS interface. Furthermore, $Li^+$ migration in the cell can be reversed by using ultraviolet (UV) light to extract valence electrons from the surface and create a driving force for $Li^+$ migration away from the surface back toward the other electrode. Our results reveal a chemically layered SEI configuration that evolves during cycling. Furthermore, the operando XPS (opXPS) measurements directly reveal the overpotentials of individual phases in the SEI, providing clues as to which SEI components limit battery performance.

## Results and Discussion

**Electrochemical cycling and ex situ XPS analysis.** To document the composition of an electrochemically formed SEI at an LPS interface, a Cu/LPS/Li coin cell was assembled and biased to drive $Li^+$ toward the Cu electrode. After 1.5 h of galvanostatic charging at 0.17 mA/cm$^2$ (Fig. 1a) the coin cell was disassembled in a glovebox and transferred through an interconnected ultrahigh-vacuum transfer system to an XPS system. The XPS spectra are shown Fig. 1b–e. The chemical states observed in these XPS spectra are in good overall agreement with the results from Wenzel et al.[11]. Electrochemical impedance spectroscopy (EIS) data sets acquired during the first few cycles of a sister sample, Supplementary Figure 1 (and Supplementary Note 1), demonstrate that the SEI continues to evolve during cycling, especially during the first few cycles. Therefore, understanding how these SEI components evolve during cycling and what effects they have on battery performance and stability are essential.

**Virtual electrode cycling and operando XPS.** XPS core-level spectra acquired from the initial LPS surface showed the presence of Li, S, P, C, and O. XPS depth profile data revealed that oxygen contamination (~3–7 atomic %) existed throughout the LPS samples tested. Consequently, oxygen was present on sample surfaces even after sputter cleaning, while no significant carbon contamination was observed (<0.5 atomic %). Figure 2a shows the Li 1s core level from the sputter-cleaned LPS surface. After a few minutes of charging via the virtual electrode (Fig. 2b) a clear shift in the Li 1s core-level is observed. As described in more detail in the following section, this shift results from the chemical transformation of LPS into SEI phases at the analysis surface, driven by $Li^+$ migration. As charging continues (>5 h in Fig. 2c) metallic Li appears at the surface, as evidenced by the appearance of a low-BE feature in the spectra.

For discharge, as shown in Fig. 2d, exposure to UV photons photoionizes metallic Li according to:

$$Li^0 \xrightarrow{h\nu} Li^+ + e^-_{\overline{PE}}, \qquad (1)$$

where $e^-_{\overline{PE}}$ denotes the outgoing photoelectron. The resulting accumulation of positive charge drives $Li^+$ away from the free surface and toward the opposite electrode so that metallic Li is stripped and the SEI is uncovered (Fig. 2e). As the SEI is

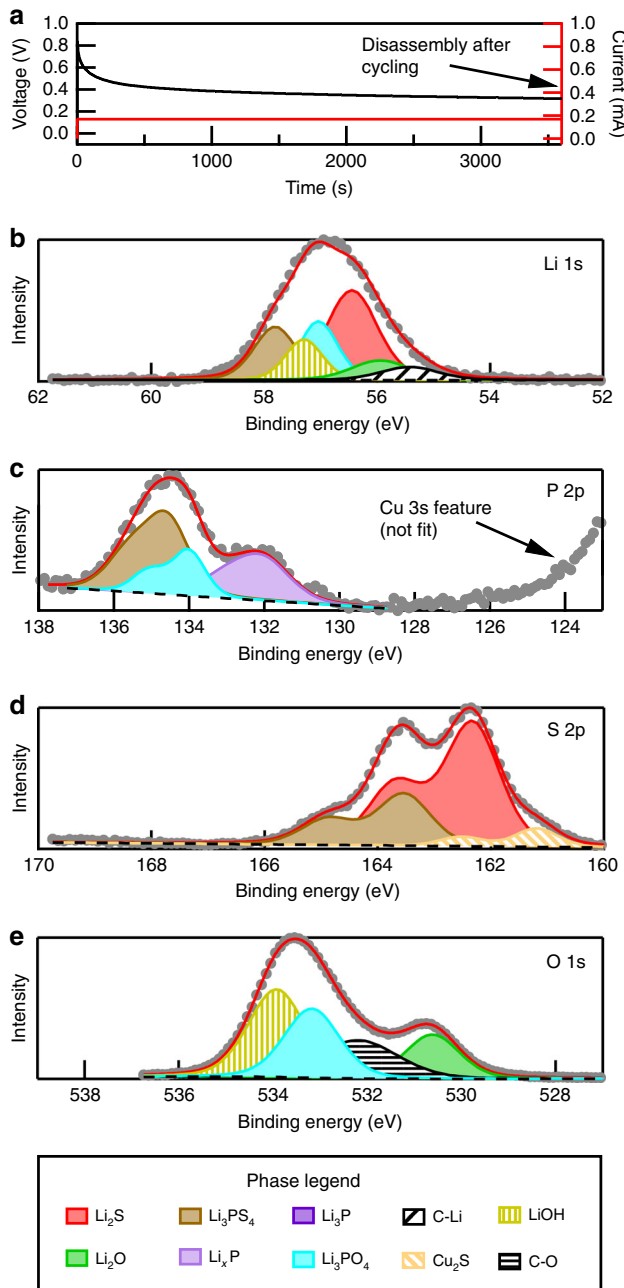

**Fig. 1** Ex situ XPS spectral decomposition of Li/LPS SEI. **a** Galvanostatic voltage profile of Cu/LPS/Li coin cell during initial charging. The cell was disassembled after 1.5 h of Li deposition onto the Cu electrode. Ex situ XPS spectra are shown for the Li 1s (**b**), P 2p (**c**), S 2p (**d**), and O 1s (**e**) core levels

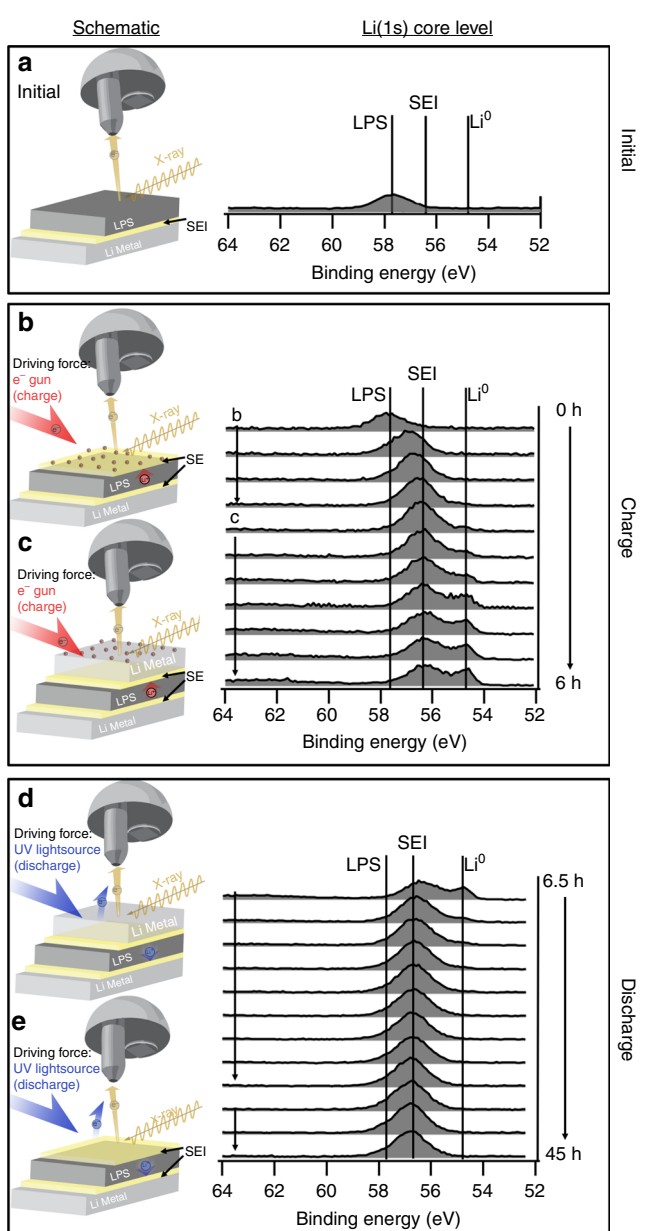

**Fig. 2** opXPS schematic and Li 1s core level evolution. **a** Clean LPS surface, **b**, **c** SEI formation during charging, and **d**, **e** SEI evolution during discharging

uncovered, spectra similar to those in Fig. 2b prior to the onset of $Li^0$ plating are observed. Eventually minimal further changes are observed in the Li 1s region, but it is noteworthy that the spectrum never returns to the initial state, indicating as expected that some SEI components are irreversible.

To provide chemical-state information related to SEI formation and evolution at the LPS/Li interface, the Li 1s and other core-level spectra shown in Fig. 3 have been decomposed into contributions from individual SEI phases. Supplementary Table 1 (and Supplementary Note 2) describes the fitting constraints used to obtain the information detailed below. As seen in Fig. 3a, the initial LPS surface is comprised almost entirely of documented

LPS functional groups[11], except for a small amount of oxygen contamination. XPS quantification using tabulated elemental XPS sensitivity factors indicates that the $O/(S + O)$ ratio is 0.14. We speculate that initially this oxygen exists primarily as isoelectronic substitutional defects on the LPS anion sublattice, $O_S$ (brown shaded traces in Fig. 3), but cannot rule out contributions from phase-separated $Li_3PO_4$ (LPO) domains. As will be discussed, oxygen-containing compounds are non-negligible components of the Li/LPS SEI. Also, recent studies have suggested that substitutional oxygen defects in LPS might enhance ionic conductivity and/or improve SEI properties[21–24]. For these reasons it is important to understand the role of oxygen in the formation and evolution of the SEI.

After 0.5 h of charging (see Fig. 3b) significant changes at the surface occurred as evidenced by the appearance of new chemical states in both the S 2p and P 2p core levels. These additional peaks indicate transformation of the P and S species at the surface

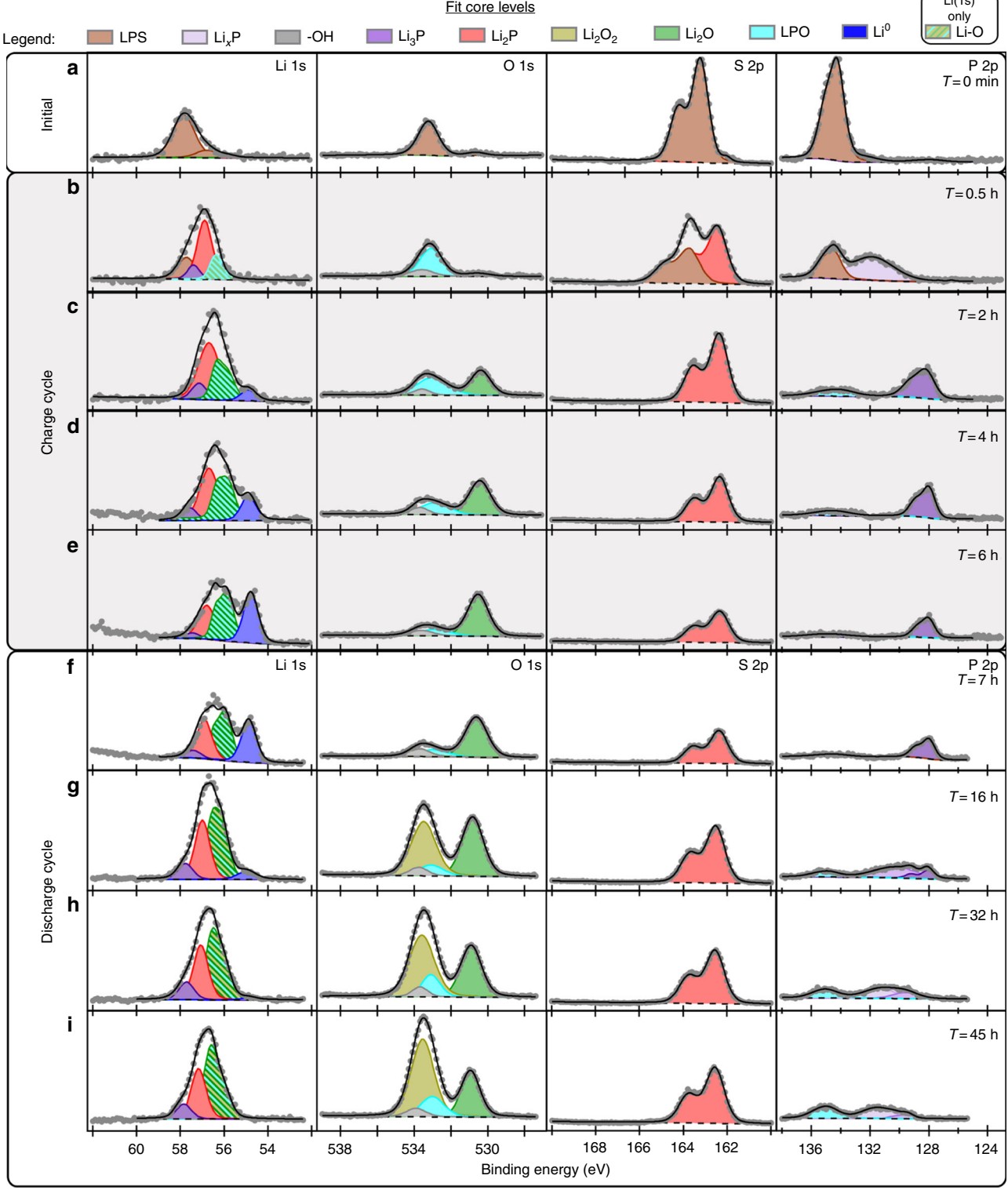

**Fig. 3** opXPS spectral decomposition of Li/LPS SEI for the first charge–discharge cycle. XPS spectra showing curve-fitting results for: **a** spectra from the sputter-cleaned LPS surface; **b**–**e** four sets of spectra during the charging step; and **f**–**i** four sets of spectra during the discharge half cycle. Due to existence of multiple O-containing species that can be fit in the Li 1s spectra, a single peak representing the summation of O-containing Li 1s components is shown with crosshatched colors of the various contributing components. For both the S 2p and P 2p core levels the 2p$_{3/2}$ and 2p$_{1/2}$ peaks have been combined into single lineshapes for simplicity. However, it should be noted that the appropriate intensity-ratio and BE separation constraints were applied during curve fitting of these features

from the original LPS functional groups (i.e., Li-S-P, P-S-P, and P = S) to $Li_2S$ (shaded red) and Li-deficient Li-P functionalities ($Li_{3-x}P$; shaded light purple). During this initial phase of charging, a slight shift to lower BE was also observed for the O 1s peak. This new BE position for the O 1s closely matches the baseline spectra acquired from $Li_3PO_4$ (LPO; shaded light blue in Fig. 3b). Therefore, we tentatively conclude that phase segregation $[4Li_3PO_xS_{4-x} \rightarrow (4-x)Li_3PS_4 + xLi_3PO_4]$ begins to occur at the surface during the charging step. It is also reasonable to conclude that after 0.5 h without sputtering, a submonolayer coverage of –OH groups exist on the surface (shaded gray in Fig. 3b), as a result of trace residual $H_2O$ vapor in the analysis chamber.

After 2 h of charging (Fig. 3c), the LPS functional groups are completely attenuated. At the same time, $Li_{3-x}P$ transforms to $Li_3P$ (shaded dark purple), $Li_2S$ increases slightly, dramatic changes in the O 1s spectra are observed, and a small amount of metallic Li ($Li^0$; shaded dark blue) appears. The striking changes in the O 1s spectra can be summarized by three observations: (i) a slight increase in the total intensity of the O 1s spectra; (ii) the appearance of a new low BE feature, assigned as $Li_2O$ (shaded green); and (iii) a decrease in signal from the $Li_3PO_4$ (LPO) phase. The origin of the slight increase in oxygen content is somewhat ambiguous, as it is possible that $Li^0$ can act as a getter material for oxygen and water residual gases in the chamber[25, 26]. However, the relatively small changes in the O 1s spectra throughout the first 0.5 h indicate that a significant amount of oxygen originated from within the sample itself rather than from background chamber contamination, as discussed in more detail later. Most importantly these results demonstrate that oxygen is an important constituent of the Li/LPS SEI, because it is difficult or impractical to remove all oxygen from the bulk and surface of these highly reactive materials during synthesis, processing, and cell fabrication. At this point during the first charge half cycle, the net charge is similar to that for the ex situ XPS analysis illustrated in Fig. 1. To demonstrate the similarities between these spectra, the XPS spectra for the cycled coin-cell and virtual-electrode samples are plotted together in Supplementary Figure 2 (also see Supplementary Note 3). The remarkable agreement between these spectra supports the validity of the virtual-electrode operando approach as a method for probing SEI evolution during electrochemical cycling. However, the comparisons of these spectra also reveal the appearance of an additional feature in the S 2p spectra for the cycled coin-cell sample that is attributed to the presents of $Cu_2S$. We conclude this phase results from side reactions between the LPS and Cu electrode.

As charging continued beyond 2 h the $Li^0$ feature increased in intensity as Li metal began to accumulate at the surface (Fig. 3d, e). This also led to attenuation of the $Li_2S$ and $Li_3P$ signals (a consequence of the limited information depth of XPS measurements, ~5–10 nm). Due to disagreements in the literature on the reported BE value for $Li^0$, a sputter-cleaned Li metal reference sample is shown in the Supplementary Information as a baseline reference (Supplementary Figure 3 and Supplementary Note 4). After charging for 6 h, a 0.5 h rest period elapsed prior to initiation of the discharge cycle.

During the initial minutes of discharging the only significant spectral change was a decrease in the amount of Li metal present at the surface and a slight transition from low-BE O 1s functional groups to high-BE species (Fig. 3f). After the majority of metallic Li had been stripped from the analysis surface (16 h, Fig. 3g), more pronounced changes in the SEI were observed for all core levels. For the S 2p core level, the $Li_2S$ peak became more intense, but the BE remained relatively unchanged. An overall increase in P 2p intensity was also observed, however the $Li_3P$ peak decreased in intensity as $Li_{3-x}P$ peaks appeared, indicating at least partial

redox activity of the Li-P phases in the SEI under Li-deficient conditions. Similarly, the $Li_2O$ component of the SEI also appears redox active, as the low BE portion of the O 1s spectra was converted into higher BE species. While there are several possible mechanisms for this conversion, perhaps the most likely is a reaction between Li-P and $Li_2O$ where $Li_2O_2$ (shaded yellow) and $Li_3PO_4$ (shaded light blue) are formed. We speculate that UV exposure photoionizes $Li_3P$ according to Eq. (2), leading to electrochemical reduction and generation of $Li^+$ according to Eq. (3). The Li-deficient Li-P phase ($Li_{3-x}P$) then chemically reacts with $Li_2O$ to form $Li_2O_2$ and $Li_3PO_4$ according to Eq. (4). Both Eqs. (3) and (4) are accompanied by migration of $Li^+$ out of the SEI through the LPS. (For Eq. (4), additional photoionization events according to Eq. (1) also occur.) The net reaction is shown in Eq. (5).

$$Li_3P \xrightarrow{h\nu} (Li_3P)^+ + e_{\overline{PE}} \tag{2}$$

$$(Li_3P)^+ \rightarrow Li_{3-x}P + xLi^+ \tag{3}$$

$$6Li_2O + Li_{3-x}P \rightarrow Li_2O_2 + Li_3PO_4 + (10-x)Li^0 \tag{4}$$

$$6Li_2O + Li_3P \rightarrow Li_2O_2 + Li_3PO_4 + 10Li^0 \tag{5}$$

Upon continued discharge (~32 h; Fig. 3h), the $Li_2O_2$ component remains relatively constant while the $Li_3PO_4$ phase grows. This coincides with a complete transformation of the $Li_3P$ into $Li_{3-x}P$, consistent with the proposed mechanism. The decrease in $Li_3P$ also demonstrates that $Li_3P$ is a redox-active component of the SEI that can be delithiated during discharge under Li-deficient conditions. By comparison, the $Li_2S$ peak appears almost completely unchanged from Fig. 3g, again indicating it is a stable component of the SEI. By 45 h of discharge (Fig. 3i) the $Li^0$ is completely gone and no other significant changes are observed.

**Evolution of SEI phase composition during cycling.** In addition to the observed chemical-state changes, the order of appearance and attenuation during charging (or intensification during discharging) of SEI phases reveals an apparent layered configuration of the SEI. We also note that the underlying LPS composition and/or SEI formation mechanisms are likely laterally heterogeneous. Consequently, spatial heterogeneities probably also exist within the SEI. Therefore, when referring to a 'layered config- uration' we do not intend to convey that the SEI is uniform and homogeneous. Rather, this terminology indicates that certain phases are more likely to form after (and on top of) other phases are already present at the interface.

As illustrated in Fig. 4, changes in the percentage phase compositions during the charge half cycle indicate that the general trend in the SEI layered structure (top to bottom) is $Li^0$/ Li-O/Li-P/$Li_2S$/LPS. Before charging, ~5% oxygen was observed in the LPS sample. After a short period of charging the amount of oxygen present in the sample remained nearly constant, whereas the $Li_{3-x}P$ and $Li_2S$ phase contents increased dramatically, from $0 \rightarrow 12$ atm% and $1 \rightarrow 50$ atm%, respectively. This indicates that initially the LPS surface reacts with the incoming flux of $Li^+$ at the surface and decomposes into $Li_{3-x}P$ and $Li_2S$. From Fig. 4, $Li_2S$ formation appears to stop first, as the signal begins to attenuate after approximately 2 h. The P containing species ($Li_3P + Li_{3-x}P$) exhibit a slight increase in total concentration until around ~3.5 h when the $Li_3P$ phase begins to attenuate. Subsequently, the $Li_2O$ signal becomes dominant, and the other

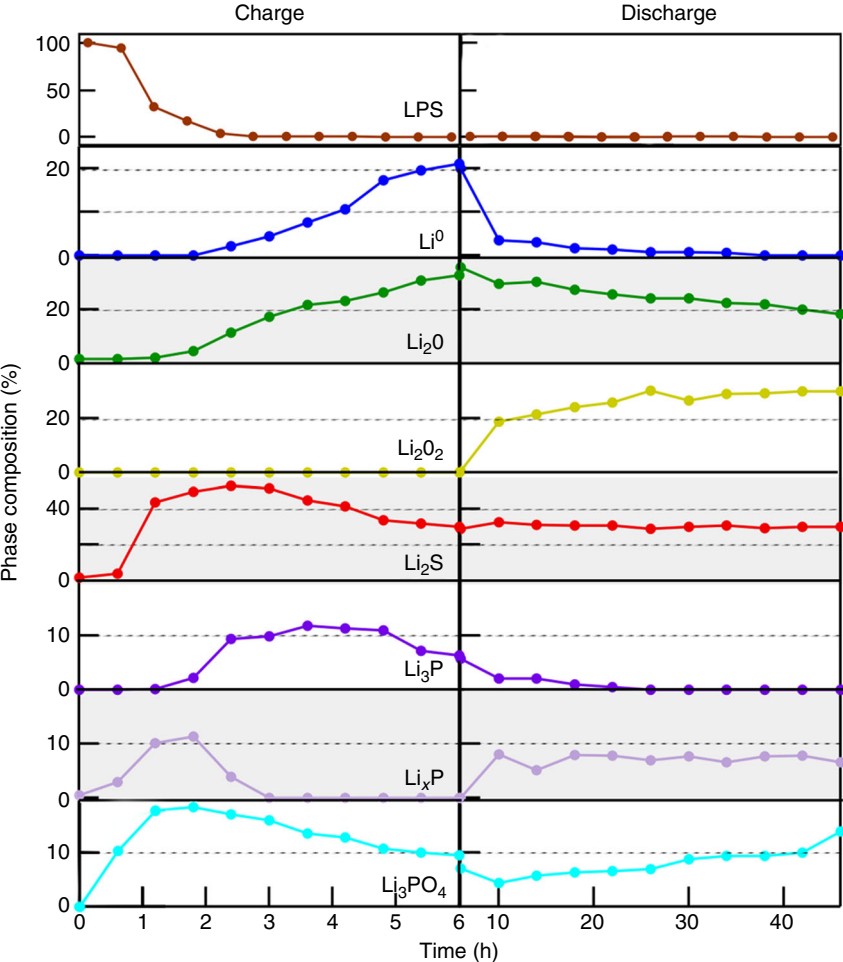

**Fig. 4** Evolution of SEI phase composition. Phase composition as a function of time for both the charge and the discharge half-cycles. Data from this graph clearly show the redox activity of the $Li_2O$ and $Li_3P$ components of the SEI during the discharge half-cycle

signals attenuate, indicating that $Li_2O$ forms on top of the $Li_2S$/$Li_3P$ layers. Eventually $Li^0$ appears at the surface, and near the end of the charge step the increase in $Li_2O$ levels off as Li metal begins to attenuate all signals from the SEI (Supplementary Figure 4 and Supplementary Note 5).

After the 0.5 h rest period, the virtual electrode polarity was reversed, and within 4 h (10 h total), significant changes in phase composition were observed. This effect was not seen in control experiments in the absence of UV light bias (Supplementary Figures 5 and 6 and Supplementary Note 6 and 7). During the discharge, as $Li^0$ was removed from the surface, the total peak intensities (Fig. 3) of O, S, and P species on the surface increase, consistent with an uncovering of the SEI. While all elements in the SEI increase in concentration, both the $Li_2O$ and $Li_3P$ phases decrease in concentration, as other Li-deficient phases appear, namely $Li_2O_2$/$Li_3PO_4$ and $Li_{3-x}P$, respectively. This highlights that some phases within the SEI are redox active, and more specifically the appearance of Li-deficient phases indicates that Li can be removed from some SEI phases (e.g., $Li_3P$ and $Li_2O$) during the discharge process. In contrast, after 4 h of discharging the $Li_2S$ phase reaches a maximum concentration, and no BE shifts occur, nor do additional S-containing phases appear. For the remainder of the discharge half cycle, this phase remains stable, indicating that $Li_2S$ is not delithiated, at least under the modest potentials applied during discharge in these experiments.

After ~20 h, no $Li_3P$ remained on the surface and the amount of $Li_2O$ had been substantially reduced. However, the amounts of

P, S, and O on the surface did not decrease compared to times after ~4 h of discharge (10 h total). The absence of substantial changes in the near-surface distribution of anion species within the SEI indicates that while some SEI components form irreversibly (e.g., $Li_2S$ and $Li_3PO_4$), others remain redox active and can be partially delithiated (e.g., $2Li_2O \leftrightarrow Li_2O_2 + 2Li^+ + 2e^-$). It should be noted that even after no further changes in the SEI were observed, $Li^+$ must have continued to flow away from the surface and through the LPS since the sample did not exhibit extreme BE shifts due to surface charging. Therefore, we speculate that this current flow is associated with ongoing compositional and chemical-state changes deeper within the SEI, outside of the XPS information depth.

**Overpotential measurements across individual SEI components.** In addition to the compositional and chemical-state changes described above, BE shifts that result from sample biasing can be related to cell polarization (described in the experimental section). By combining data from these cell polarization measurements with the proposed layered configuration of the SEI, the overpotential ($\eta$) associated with each SEI phase can be estimated. This is calculated according to Eq. (6) by subtracting the cell polarization measured at one layer, $\phi_A$, from the cell polarization observed for an adjacent layer, $\phi_B$, (where layer B is on top of layer A) according to:

$$\eta_B = \phi_B - \phi_A. \qquad (6)$$

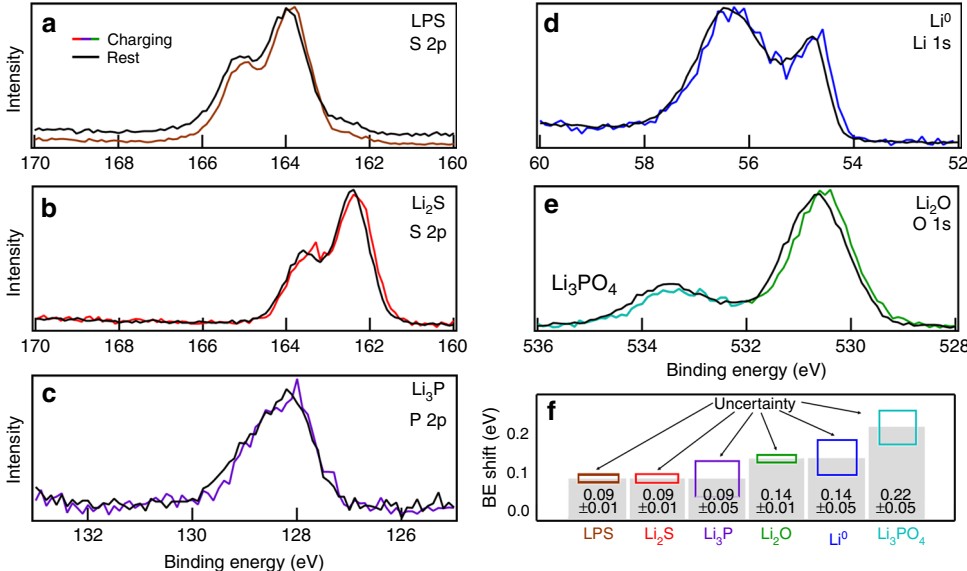

**Fig. 5** opXPS determination of chemically resolved cell polarization/overpotential. XPS spectra showing chemically resolved BE shifts caused by applied current bias **a** before charging (LPS/Li$_{foil}$); and **b**–**e** after the charging half-cycle (Li/LPS/Li$_{foil}$). **f** Summarizes the current-bias-induced BE shifts extracted by peak fitting for each of the observed SEI phases. Individual overpotential losses associated with each phase can be determined from these shifts

Figure 5 shows XPS spectra from a sample with and without bias. These data were acquired at two different points during charging: before cycling (to measure the cell polarization of the initial LPS/Li$_{foil}$ sample; Fig. 5a), and after the charge step (to measure cell polarization values across various SEI phases; Fig. 5b–e). From these spectra, BE shifts have been extracted via curve fitting and are summarized in Fig. 5f. This analysis shows that the net cell polarization across the LPS/Li$_{foil}$ stack is $90 \pm 10$ mV (brown) at 0.17 mA/cm$^2$. Assuming that negligible impedance changes occur across this portion of the cell during the first charge step, overpotentials across individual SEI phases can be extracted. Comparing the voltage drops of both the Li$_2$S (red; $90 \pm 10$ mV) and the Li$_3$P (purple; $90 \pm 50$ mV) to that of the initial LPS/Li$_{foil}$ reveals there are minimal contributions to cell overpotential from these SEI components. This indicates that these phases are either extremely thin or have good ionic conductivity. By comparison, the cell polarization associated with Li$_2$O (green) is much higher, $140 \pm 10$ mV, indicating that a significant overpotential drop occurs across this layer ($\eta_{Li_2O} = 50$ mV). For Li$^0$, the same cell polarization as Li$_2$O is observed (blue; $140 \pm 50$ mV). This is consistent with Li$^0$ plating on top of Li$_2$O, supporting the conclusion that oxygen observed in SEI species originates within the LPS SSE. Perhaps most importantly, the BE shift for the peaks associated with Li$_3$PO$_4$ show the largest total cell polarization $220 \pm 50$ mV. This suggests that Li$_3$PO$_4$ either forms on top of Li$^0$ (unlikely), or that Li$_3$PO$_4$ exhibits such poor ionic conductivity that it acts as a blocking layer for Li$^+$ transport. It is also noteworthy that local cell polarizations determined for Li$_2$O and Li$^0$ (140 mV) closely match the value determined from a cycled symmetric coin cell (Supplementary Figure 7 and Supplementary Note 8), again indicating that the Li$_2$O layer is in the primary conductive pathway through the SEI. The overpotentials for each phase are summarized at the end of the next section.

**Observation of SEI non-uniformity via in situ AES mapping.** To provide further evidence that Li$_3$PO$_4$ inhibits Li$^+$ migration, the virtual electrode concept was used in combination with AES mapping to provide two-dimensional compositional information related to SEI formation. These results, shown in Fig. 6a–d (and Supplementary Movie 1), reveal the Li/LPS interface evolves

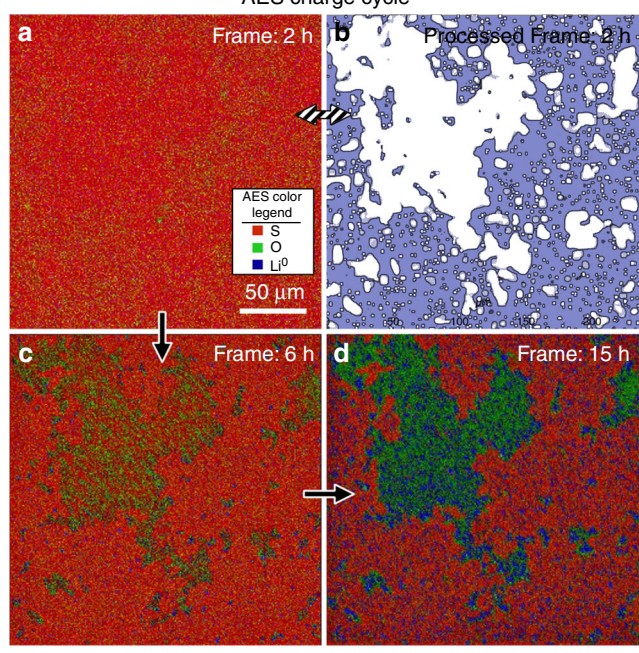

**Fig. 6** In situ AES mapping of SEI heterogeneity. **a** shows an AES mapping image of the LPS SEI after 2 h of charging. **b** is a threshold-filter processed version of the same 2 h AES map, where white areas highlight little or no oxygen and blue areas represent higher oxygen concentrations, likely in the form of Li$_3$PO$_4$. **c**, **d** show the SEI after 6 and 15 h of charging, respectively

heterogeneously during the charge cycle. Careful inspection of Fig. 6a, representing the initial surface, reveals a heterogeneous distribution of oxygen. To highlight these subtle differences, the AES map in Fig. 6a was processed via a series of threshold filters, to show areas of higher vs. lower oxygen content (light blue vs. white, respectively) in Fig. 6b. Comparison of this processed image to the final AES map in Fig. 6d reveals a clear correlation between the SEI where Li$_2$O (green) and Li$^0$ (blue) appear and the white areas in Fig. 6b. This observation is consistent with the

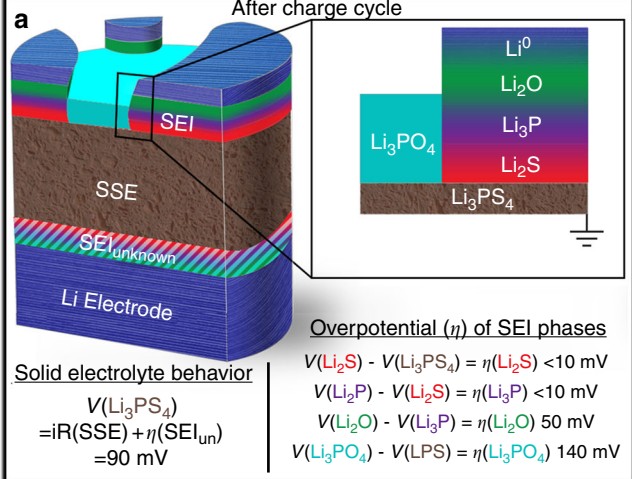

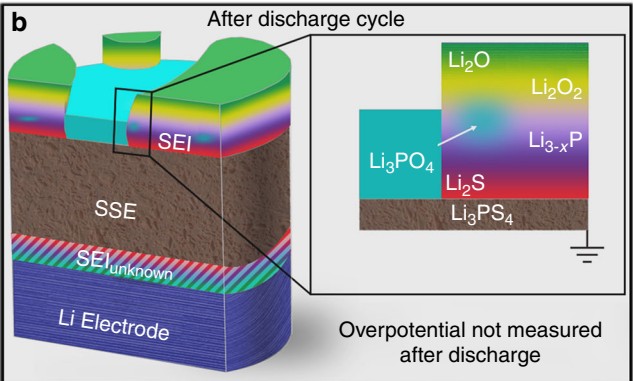

**Fig. 7** Summary of SEI composition and morphology. A schematic is shown of the three-dimensional SEI structure after the charge (**a**) and discharge (**b**) step. A summary of the individual SEI overpotentials measured after the charging step is shown in (**a**)

hypothesis that oxygen contamination in LPS initially leads to $Li_3PO_4$ phase segregation, but also shows that oxygen content is initially non-uniform, and consequently $Li_3PO_4$ forms predominantly in oxygen-rich areas. Subsequently the $Li_3PO_4$ inhibits $Li^+$ transport in those areas, so that other SEI phases such as $Li_2O$ form in complementary regions of the surface. This underscores the importance of controlling and understanding oxygen contamination and defects in LPS SSE's. Furthermore, the strong correlation between oxygen-rich areas of the SEI after charging (presumed to be $Li_2O$) and the appearance of $Li^0$ in Fig. 6 (and Supplementary Movie 1) provides further evidence for both the lateral and layered structure of the SEI.

To illustrate these conclusions graphically, Fig. 7 shows the proposed layered configuration after both charging and discharging. The phrase 'layered configuration' is not intended to convey a homogeneous structure across the entire interface, rather this term is used to describe the propensity of certain phases to form before others. Therefore, the schematic shown in Fig. 7 is the most likely structure of the SEI, although other configurations might exist at various points across the surface. Figure 7a also demonstrates the measured overpotentials across the individual SEI phases. Based on statistical analysis of local cell polarization values extracted from BE shifts (see Supplementary Figure 8, Supplementary Table 2 and Supplementary Notes 9 and 10), we can say with 99.8% certainty that $Li_3P$ has a lower local cell polarization than $Li_2O$. Similarly, it can be stated with 99.6% certainty that $Li_3PO_4$ has a higher local cell polarization than $Li^0$. The AES mapping results provide additional support for these

conclusions. Unfortunately, it is not possible to say with certainty whether $Li_3PO_4$ is above or adjacent to the $Li_2S$, $Li_3P$, or $Li_2O$ phases. However, based on the combination of XPS and AES data it seems most likely that $Li_3PO_4$ initially forms directly on the LPS.

In this work we have developed and implemented the virtual electrode approach for cycling solid-electrolyte-based batteries while monitoring interface composition and properties with XPS or AES. These techniques enable operando and in situ measurements of SEI formation and electrochemical evolution during charge-discharge cycles. A primary finding from this study is that the evolution of phase compositions within the XPS information depth provides evidence for a layered configuration within the Li/LPS SEI. Specifically, the opXPS measurements reveal that $Li_2S$ and Li-P phases form first, followed by Li-O phases, and finally $Li^0$. OpXPS results also reveal that $Li_3P$ and $Li_2O$ are redox-active SEI phases, as indicated by concentration and chemical-state changes in the P 2p and O 1s core levels during discharge, consistent with transformation of these phases into Li-deficient compositions (i.e. $Li_2O \rightarrow Li_2O_2$ and $Li_3P \rightarrow Li_{3-x}P$). On the other hand, $Li_2S$ is a stable SEI phase that does not evolve further during cycling, at least for the modest electrochemical conditions tested herein.

Our results also demonstrate that controlled surface charging using the virtual electrode approach enables overpotential losses across individual SEI phases to be directly quantified. Relative BE shifts in XPS spectra during biased and non-biased conditions allow for individual measurements of overpotentials, which can be assigned to the specific SEI phases. In the case of LPS solid electrolytes, results from this analysis show that the $Li_2S$ and Li-P SEI components cause negligible polarization losses. On the other hand, the oxygen-containing components exhibit much larger polarization losses. OpXPS results coupled with in situ AES measurements reveal that $Li_3PO_4$ inhibits $Li^+$ migration. The $Li_3PO_4$ SEI phase forms initially through phase segregation of oxygen-contaminated LPS ($Li_3PO_xS_{4-x}$), and subsequently grows during discharge as a consequence of reactions between the delithiated $Li_{3-x}P$ and $Li_2O_2$ phases. Of all SEI phases observed, $Li_3PO_4$ exhibits the largest overpotential, indicating that $Li_3PO_4$ hinders $Li^+$ migration and therefore is likely detrimental to battery performance, cycling stability and the uniform deposition of Li. To the best of our knowledge, these opXPS measurements represent the first chemically resolved overpotentials that have been measured across individual SEI phases.

## Methods

**LPS synthesis.** The LPS electrolyte was prepared with a molar ratio of 77.5% $Li_2S$ (99.9% Alfa-Aesar) and 22.5% $P_2S_5$. (99% Sigma-Aldrich). All material handling and electrode fabrication occurred in an argon environment. The material was planetary milled for 20 h in 500 mL stainless steel jars. LPS samples for opXPS experiments were fabricated by compressing 200 mg of powder at 75 MPa into 13-mm diameter pellets approximately 1.2 mm thick. This step was followed by application of a lithium metal-foil electrode on a single side at 15 MPa. Samples used during galvanostatic cycling and AC impedance testing consisted of one lithium electrode and one copper electrode pressed onto the LPS. In addition, one galvanostatic cycling data set shown in the SI was acquired from a symmetric-cell-configuration sample consisting of lithium metal electrodes on both sides of the LPS.

**XPS analysis.** OpXPS measurements were performed in a Physical Electronics 5600 photoelectron spectrometer under ultrahigh vacuum (UHV) conditions, with typical pressures below $5 \times 10^{-10}$ Torr. The Al Kα X-ray monochromator was operated with an anode power of 350 W. The sample surface normal was oriented at 45° to both the X-ray source and photoelectron spectrometer. The spectrometer pass energy was set to 29.35 eV. Binding-energy (BE) calibrations on sputter-cleaned metal foils followed values reported by Seah for Au $4f_{7/2}$, Ag $3d_{5/2}$, and Cu $2p_{3/2}$ core-levels[27]. Curve fitting and data processing was performed using Igor Pro with a custom program adapted from Schmid et al.[28].

**Auger Electron Spectroscopy (AES) analysis**. Real-time in situ AES measurements were performed using a Physical Electronics 670 system, under beam energy of 5 kV, with 20-nA beam current. Typical pressures ranged from $3 \times 10^{-10}$ to $1 \times 10^{-9}$ Torr.

**Electrochemical operando XPS**. Cohen and coworkers have previously used electron-gun induced, controlled surface charging to develop a non-destructive XPS depth-profiling method that was applied to studies of metal–organic coordinated multilayers[29]. Subsequently, they used a similar approach to probe alkali metal deintercalation from inorganic fullerene-like materials[30]. In the present work, we use a similar approach, where an applied current bias enables electrochemical cycling in a battery device structure by alternating between an electron source and a photon source. We refer to this method of supplying charge to the surface as the 'virtual electrode'.

**Charge half-cycle**. In the 'virtual-electrode' approach developed in this work, a Physical Electronics model 04-090 electron flood gun was used to supply electrons to the surface during the charge half-cycle. In a typical XPS application this electron gun provides a flux of low-energy electrons that neutralize surface charge on insulating samples. Generally, higher electron kinetic energies are associated with overall higher beam currents. Experiments were performed to characterize the relative tradeoffs between electron kinetic energy vs. net electron flux at the surface. Based on these measurements, the kinetic energy was set to 11.5 eV (the maximum for this electron gun), which produced an incident current density, measured with a Faraday cup, of ~0.17 mA/cm$^2$. In virtual electrode opXPS experiments, the electron flux to the surface results in a net negative surface charge that induces an opposing Li$^+$ ion migration flux. Equilibrium conditions are established as soon as the built-up negative surface charge provides a sufficient driving force for migration of Li+ through the LPS/Li$_{foil}$ sample, such that the two fluxes exactly balance. The net negative charge at the surface manifests as XPS BE shifts, $\Delta$BE, equivalent to the voltage bias across the sample: $V_{bias} = \Delta BE/q$, where $q$ is the electronic charge. In the present study, the timescale for equilibration of electronic and ionic fluxes (as revealed by observed BE shifts) is short, on the order of <1 s. Importantly, the conclusion that net electronic and ionic current densities are equal in magnitude depends on the assumption that the LPS electronic conductivity is negligible. It should also be noted that the net electron current density at the electrode during opXPS could be lower than the incident electron flux, due to the effects of elastic and inelastic backscattering and surface charging effects. For example, the net flux into a metallic Li foil sample using the same flood gun settings produced a net electron current ~0.12 mA/cm$^2$, and we expect that the ratio of incident electron flux to net current density is material dependent to some extent. OpXPS control experiments on LPS samples (Supplementary Figure 9 and Supplementary Note 11) using a lower electron kinetic energy (2.2 eV) did not show any significant differences aside from slower rates of Li$^+$ migration associated with lower incident electron fluxes.

**Discharge half-cycle**. To implement the virtual electrode concept during the discharge half-cycle, net positive surface charge was generated through photo-emission using UV photons ($\lambda = 405$ nm, $hv = 3.06$ eV) supplied by an LED (Sensor Electronic Technology, Inc.). This positive surface charge provided the driving force for Li$^+$ migration away from the free surface and into the LPS sample. The LED was mounted outside the UHV chamber and light was focused through a viewport onto the sample surface. Due to the very low kinetic energies of photo-electrons created by the UV LED, a negative voltage bias (~45 V) was applied between ground and the Li metal-foil electrode. This arrangement biases the entire LPS/Li$_{foil}$ sample relative to ground potential in the vacuum chamber, which accelerates low-energy photoelectrons away from the sample surface, thereby maximizing ionic current density. With UV illumination and applied voltage bias, net photoemission currents ~150 nA were measured, indicating that a sufficient electrochemical driving force is applied by this method to drive Li$^+$ ion migration away from the surface and through the LPS. Based on calibration measurements using an area-defining aperture, the estimated ionic current density during discharge is ~0.01–0.02 mA/cm$^2$.

**Battery test structure and virtual electrode**. Figure 8 shows a schematic of the opXPS experiment. The sample consists of LPS mechanically pressed onto a Li metal-foil electrode. Based on measurements by Wenzel et al.[11], when the Li metal foil is brought into contact with LPS an SEI forms creating a LPS/SEI/Li$_{foil}$ structure. (In this paper we use LPS/Li$_{foil}$ and LPS/SEI/Li$_{foil}$ interchangeably, recognizing that an SEI always present at the rear LPS/Li$_{foil}$ interface.) During measurements the Li$_{foil}$ electrode is in electrical contact with the grounded or voltage-biased XPS sample stage. In this configuration, the bare LPS surface is initially exposed to only the X-ray analysis beam, in order to observe initial composition and chemical states. To drive Li$^+$ through the LPS toward the surface (Fig. 8a), the virtual electrode is applied by exposing the analysis surface to a flux of low-energy electrons. After charging a symmetric cell configuration is formed (i.e., Li/SEI/LPS/SEI/Li$_{foil}$).

To investigate the dynamic nature of the SEI during cycling, the virtual electrode concept was also used to supply net positive charge to the exposed

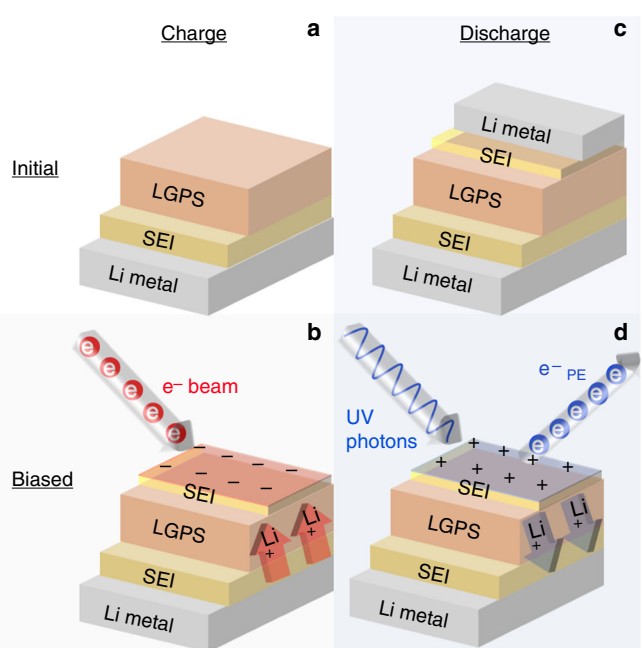

**Fig. 8** Schematic of opXPS virtual electrode method. Schematic showing **a** the initial structure of the LPS/Li$_{foil}$, **b** the structure part of the way through the charge step, **c** the structure after the charge cycle is complete before right discharge and **d** the Li/LPS/Li$_{foil}$ during the discharge step

analysis surface. This was accomplished by applying UV photons to the top surface (Fig. 8d), which eject valence electrons from phases with ionization potentials less than the photon energy. For the materials in the present study, this creates an equilibrium accumulation of excess Li$^+$ ions near the surface, which in turn drives Li$^+$ migration away from the analysis surface, through the LPS, back to the Li metal-foil electrode. This process is referred to as the discharge half cycle. After discharge the cell configuration is SEI/LPS/SEI/Li$_{foil}$.

It is important to note that the magnitude of the current density associated with the X-ray photoemission process used to generate XPS spectra is typically much smaller than those associated with either the electron flood gun or the UV LED light source, which together comprise the so-called virtual electrode. Nevertheless, in control experiments where high X-ray fluxes were supplied via a standard (non-monochromatic) Al anode X-ray source, we observed effects of Li$^+$ migration caused by the photoelectron flux associated with the XPS measurement. Since X-ray photoemission extracts core-level electrons as well as valence electrons from all atomic species in a sample, it is possible that X-ray exposure can drive non-equilibrium electrochemical processes, which might more commonly be referred to as beam damage. Therefore, steps were taken to minimize net X-ray exposure during the opXPS measurements to levels well below beam damage thresholds determined in control experiments.

**Operando XPS cell polarization/overpotential measurements**. In this study local cell polarization, $\phi$, refers to the net potential difference between the Li$_{foil}$ electrode and an arbitrary point within the cell. Varying conventions exist for the definition of overpotential. To avoid confusion, herein local overpotential refers to the difference between local cell polarization and the sum of the equilibrium (thermodynamic) reduction/oxidation potentials combined with voltage losses associated with mass transport through the electrolyte. Therefore, this definition of local overpotential only includes voltage loss contributions associated with activation and transport limitations within the SEI. Information relating to both local cell polarizations and overpotentials can be extracted from opXPS measurements by comparing XPS core-level BE positions as a function of bias conditions. During charge and discharge half-cycles, current applied by the virtual electrode causes the cell to self-bias, creating a potential drop across the cell. In the case of an SEI with a layered structure, core levels associated with a particular SEI phase shift by the local cell polarization measured between ground potential and that phase. Therefore, for the initial LPS/SEI/Li$_{foil}$ sample, the change in core-level BEs between the biased and unbiased condition is equal to the overpotential across the buried SEI added to the internal resistance voltage drop across the electrolyte. Similarly, by observing the biased vs. unbiased core-level BE shifts at the end of the charge half-cycle, it is possible to determine local overpotentials associated with individual phases within the SEI that has formed on the LPS surface.

**Real-time in situ AES mapping measurements**. The AES electron gun was used to charge the sample surface via the virtual electrode approach, and concurrently to

obtain sequential AES compositional images. The electron gun accelerating voltage and beam current were set to 5 kV and 20 nA, respectively. The ~25 nm diameter electron beam probe thereby resulted in quite large local instantaneous beam currents (>1000 A cm$^{-2}$). On the other hand, rastering the beam over a 200 μm × 250 μm ($5 \times 10^{-4}$ cm$^2$) area reduced the time-averaged current density to 0.04 mA cm$^{-2}$. Nevertheless, because charging with a rastered electron beam differs significantly from current flow in standard electrochemical cells, we describe the AES mapping measurements as 'real-time in situ' rather than 'operando'. Also, it is worth noting that in spite of the relatively high electron kinetic energies used in AES measurements, no significant beam-damage artifacts were observed.

**Data availability**. All relevant data are available from the corresponding authors upon request.

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

## Acknowledgements

This work was authored by Alliance for Sustainable Energy, LLC, the manager and operator of the National Renewable Energy Laboratory for the U.S. Department of Energy (DOE) under Contract No. DE-AC36-08GO28308. Funding provided by the Laboratory Directed Research and Development program. The views expressed in the article do not necessarily represent the views of the DOE or the U.S. Government. The U. S. Government retains and the publisher, by accepting the article for publication, acknowledges that the U.S. Government retains a nonexclusive, paid-up, irrevocable, worldwide license to publish or reproduce the published form of this work, or allow others to do so, for U.S. Government purposes. Work at the University of Colorado was supported by National Science Foundation (CBET, 1605528)

## Author contributions

G.T. supervised the project and conceived the virtual electrode approach for XPS and AES measurements. G.T., K.X.S., C.B., and S.S. conceived and planned opXPS measurements on solid electrolyte materials. K.N.W. and G.T. further developed the opXPS technique for battery analysis, performed the experiments and analyzed the data. S.E.H and S.H.L fabricated the samples and preformed the ex situ electrochemical measurements. K.N.W. and G.T. wrote the initial draft of the manuscript. All other co-authors discussed the results and contributed to the final manuscript preparation.

## Additional information

**Competing interests:** The authors declare no competing interests.

