## [Peer Review File · Nature Communications]

Reviewer #1 (Remarks to the Author):

This is a quite fine piece of work, emphasizing novel electrochemical operando XPS and AES mapping techniques have been developed and applied to probe the formation and evolution of the LPS-Li SEI during cycling

I suggest publishing the work if the following items are addressed:

(1) Figure 4 summarizes XPS spectra showing curve-fitting results. LPS and Li₂S fitting data peaks in the S 2p spectra indicate double peak. The double peaks were strange as fitting curve in the XPS analysis. Please address the correct compound in the manuscript.

(2) Figure SI1 summarized EIS spectra showing the evolution of the SEI during the first three cycles of a Cu/LPS/Li coin cell. I think the figure which understands EIS data should be written clearly in order to measure initial conductivity. Please address this point.

Reviewer #2 (Remarks to the Author):

This paper is very interesting and should be published in Nature. It is presenting an elegant experiment but of course there are details which should be considered to clarify the results more. It is clear that the main expertise is in the photoelectron spectroscopy. Since the paper is very well written it is only in the nitty gritty details where I have my few points to discuss.

In general operando is called in operando. It is latin and suggested first by a Spanish scientist.

Experimental:

The lithium metal will have a surface of compounds with a composition dependent on how it is stored. It will also directly react with the atmosphere in a glove-box. There is no analysis of the surface of the pristine lithium? It would be good to know if the oxygen is already there from the start.

A pellet of 1.2 mm for the electrolyte is extremely thick for a lithium battery. It may affect the impedance of the coin cell study.

Results:

I think the suggested way of study the SEI is very elegant. I am just worried that the spectra in the ex situ study (Figure 2) of the coin cell are very broad (several eVs) and feature-less. Can you really draw large conclusions from them? I am sceptical since the broad shape could be a result of partial charging of the surface measured rather than new compound appearing?

I wonder if your results are affected by the SEI components having a different binding energy than the bulk of the sample? Should the SEI and the bulk peaks really be treated with the same calibration or is there a dipole effect in this interface? There are discussions about this in the literature based on HAXPES or tender XPS studies.

March 20th, 2018

Dr. Adam West
Senior Editor
Nature Communications

RE: Response to reviewer comments for manuscript NCOMMS-17-29108-T entitled: *Electrochemical AES Mapping and operando XPS Measurements of SEI Formation and Evolution in Li₂S-P₂S₅ Solid-State Electrolytes* by Kevin N. Wood, K. Xerxes Steirer, Simon E. Hafner, Chunmei Ban, Shriram Santhanagopalan, Se-Hee Lee, and Glenn Teeter

Dear Dr. West,

We greatly appreciate your review and have modified our manuscript to address the helpful reviewer comments. Our responses to reviewer comments, detailing those changes, are provided below. Changes are highlighted in yellow in the revised manuscript and in the supporting information. To summarize our work, we have employed the novel concept of the ‘virtual electrode’ to enable electrochemical cycling of solid electrolyte battery systems on exposed interfaces during XPS and AES measurements. Our measurements reveal how the SEI forms and evolves during cycling, as well as individual overpotential contributions from each SEI constituents. We believe that this study provides valuable insights into SEI composition and evolution in LPS-based solid electrolytes; and the novel techniques we have developed to help researchers address the major hurdles facing next generation batteries. Thank you for your consideration of this manuscript for publication in Nature Communications.

Reviewer #1:

This is a quite fine piece of work, emphasizing novel electrochemical operando XPS and AES mapping techniques have been developed and applied to probe the formation and evolution of the LPS-Li SEI during cycling

I suggest publishing the work if the following items are addressed:

Comment 1: Figure 4 summarizes XPS spectra showing curve-fitting results. LPS and Li₂S fitting data peaks in the S 2p spectra indicate double peak. The double peaks were strange as fitting curve in the XPS analysis. Please address the correct compound in the manuscript.

Response 1: The apparent ‘double peak’ behavior arises from the 2p_{3/2} and 2p_{1/2} spin-orbit splitting for both the S 2p and P 2p core levels. This spin-orbit splitting results in two peaks for each phase that contains S or P. On the other hand, both the O-1s and Li-1s core levels are characterized by single peaks for each unique chemical state that is present. So, for example, the Li₂S has two peaks in the S 2p core level (2p_{3/2} and 2p_{1/2}), but only one peak in the Li 1s core level. For all 2p core levels, the expected intensity ratio between the 2p_{3/2} and 2p_{1/2} peaks is 2:1. Similarly, for a particular element there are characteristic energy separations between the 2p_{3/2} and 2p_{1/2} levels. Both of these constraints (intensity ratio and binding energy separation) were applied during the curve-fitting procedure to ensure rigorous and accurate results. In order to simplify the presentation of spectra in Figure 4, the peaks associated with 2p_{3/2} and 2p_{1/2} orbitals for both the S 2p and P 2p were combined into single lineshapes representing each observed core level. We appreciate the reviewers comment and understand the potential for confusion, so to clarify these points we have added the following text to the caption of Figure 4:

For both the S 2p and P 2p core levels the 2p_{3/2} and 2p_{1/2} peaks have been combined into single lineshapes for simplicity. However, it should be noted that the appropriate intensity-ratio and BE separation constraints were applied during curve fitting of these features.

Comment 2: Figure SII summarized EIS spectra showing the evolution of the SEI during the first three cycles of a Cu/LPS/Li coin cell. I think the figure which understands EIS data should be written clearly in order to measure initial conductivity. Please address this point.

Response 2: We agree that the EIS spectra should be used to track conductivity of the bulk electrolyte. These values are listed in the legend of Figure SII. To clearly show the conductivity values of the LPS pellet we have added the following text to the SI:

Fitting the component associated with the bulk of the electrolyte reveals the conductivity values for the LPS samples to be 1.61 mS/cm during first half cycle, 1.42 mS/cm after the second half cycle and 1.35 mS/cm after three full cycles. These results are summarized in the legend of Figure SII.

Reviewer #2:

This paper is very interesting and should be published in Nature. It is presenting an elegant experiment but of course there are details which should be considered to clarify the results more. It is clear that the main expertise is in the photoelectron spectroscopy. Since the paper is very well written it is only in the nitty gritty details where I have my few points to discuss.

Comment 1: In general operando is called in operando. It is latin and suggested first by a Spanish scientist.

Response 1: The authors have seen both ‘operando’ and ‘in operando’ used in the literature, and do not have a strong preference for either. Therefore, we propose to defer this decision to the Nature Communications style guide and editors.

Comment 2: The lithium metal will have a surface of compounds with a composition dependent on how it is stored. It will also directly react with the atmosphere in a glove-box. There is no analysis of the surface of the pristine lithium? It would be good to know if the oxygen is already there from the start.

Response 2: We agree with the reviewer that a metallic Li surface is extremely difficult to keep clean, whether in an Ar glovebox or under ultrahigh-vacuum (UHV) conditions. However, the authors would like to point out that in our studies, metallic Li initially exists only on the ‘bottom’ of the test structure (in the form of Li foil) as seen in Figure 1 in the main text. The Li metal foil anodes were not directly probed in the present study. The virtual electrode approach that we developed allowed us to plate nanoscale Li⁰ films on the front surface of the LPS samples, following formation and passivation of the SEI. We are aware that, given enough time, reactions with trace gases in the in the XPS chamber would eventually convert the metallic Li surface layer to compounds including Li₂O, LiOH, Li₂CO₃, etc. We discuss these issues in the SI:

“Due to the extreme reactivity of Li metal and the unavoidable presence of trace H₂O and other gases in the XPS chamber (typically < 5 × 10⁻¹⁰ Torr), it is expected that small amounts of oxygen-containing species form continuously on the exposed Li⁰ layer. On the other hand, we can estimate the minimum time for one complete monolayer (ML) of O-containing phases at the Li⁰ surface as follows. For 5×10⁻¹⁰ Torr, using H₂O as an example, we calculate an impingement rate ~6×10¹¹ molecules•cm⁻²•s⁻¹. Assuming unity sticking coefficient, one full ML (~1.3×10¹⁵ Li

sites $\cdot\text{cm}^{-2}$) of $-\text{O}$ or $-\text{OH}$ will form in ~ 1 h. For the opXPS experimental conditions tested, we estimate the Li^+ current density to be $> 0.1 \text{ mA cm}^{-2}$, corresponding to $6 \times 10^{14} \text{ Li}^+ \text{ cm}^{-2} \text{ s}^{-1}$, or $\sim 0.5 \text{ ML s}^{-1}$. Therefore, at typical opXPS experimental conditions, the Li^+ arrival rate at the surface is ~ 1000 times larger than that of residual gas molecules. On this basis we conclude that Li^0 plating occurs faster than conversion to O-containing phases by reactions with trace gases.”

In addition, to further verify that the changes in SEI composition that we observed resulted from electrochemical effects (as opposed to reactions with trace gases) we performed control experiments as described in the SI in Figs. SI5-7 and associated text. To summarize, we concluded decisively that reactions with residual gases could not explain our observations, and expect that overall these effects had negligible impacts on our experiments and conclusions.

Beyond what we report in the present study, we have also performed calibration experiments to characterize the surfaces of as-received and ‘pristine’ Li metal, the latter obtained via extensive Ar^+ -ion sputter cleaning. The characteristic XPS spectral features that we have observed from clean Li-metal surface include: a Li 1s peak at 55.0 eV; a plasmon-loss feature at ~ 62.5 eV; and a valence-band spectrum characterized by a metallic Fermi edge at 0.0 eV. These baseline measurements fully support our interpretation that metallic Li does in fact appear on the surface of LPS samples after formation of the SEI during charging. To address the reviewers concerns and make these points more concrete, the following reference spectra have been added to the SI and referenced in the main text.

Figure SI4. Reference Li 1s and VB spectra acquired from a Li-foil sample following extensive Ar^+ -ion sputter cleaning to obtain a ‘pristine’ Li metal surface (~ 0.5 atomic % residual oxygen). Characteristic features observed for pristine Li^0 include: a Li 1s peak at 55.0 eV; a plasmon-loss feature at ~ 62.5 eV; and a valence-band spectrum characterized by a metallic Fermi edge at 0.0 eV.

Comment 3: A pellet of 1.2 mm for the electrolyte is extremely thick for a lithium battery. It may affect the impedance of the coin cell study.

Response 3: We agree with the reviewer that 1.2 mm might seem thick for a battery application, however for the opXPS experiment the thickness of the sample does not adversely affect the analysis or data interpretation. With opXPS measurements, we typically observed cell overpotential values ~ 0.2 V or less that could be attributed to the combination of impedances from the bulk electrolyte and the SEI between the Li-foil anode and the LPS pellet. The EIS-derived conductivity values shown in Figure SI1 support these observations, and indicate that the 1.2-mm thick electrolytes can provide reasonable performance.

Comment 4: I think the suggested way of study the SEI is very elegant. I am just worried that the spectra in the ex situ study (Figure 2) of the coin cell are very broad (several eVs) and feature-less. Can you really draw large conclusions from them? I am sceptical since the broad shape could be a result of partial charging of the surface measured rather than new compound appearing?

Response 4: Charging can indeed be a significant issue that affects XPS analyses of battery materials, especially on materials like solid-electrolytes that are poor electronic conductors. However, a key concept

that enabled development of the virtual electrode approach is that ionic conduction in solid-electrolyte materials coupled with intentional surface charging maintains overall charge balance while driving Li^+ migration. As noted previously, in LPS/ Li_{foil} samples we typically observed *operando* bias voltages ~ 0.2 V (corresponding to BE shifts ~ 0.2 eV) or less. When no *operando* bias was applied, on the other hand, we observed no detectable artifacts in XPS spectra that could be attributed to charging effects, such as pronounced peak asymmetries or binding-energy shifts that respond sensitively to charge neutralization. In our lab we have also recently performed systematic XPS binding-energy calibration measurements on many Li compounds phases commonly observed in SEIs, including Li^0 , Li_2O , Li_2O_2 , LiOH , Li_2CO_3 . These experiments were specifically designed to probe charging effects that commonly occur in these materials, and to develop data-analysis strategies that enable accurate phase determinations in spite of charging effects. This study is currently under review at another journal, but based on those results, as well as studies in the literature, we are very confident in the phase assignments we have extracted from XPS curve fitting.

To address the reviewers specific concern that the XPS spectra from the cycled coin cell are very broad, we have modified main text Fig. 2 and Fig. SI2 to include the full spectral decomposition (as in main text Fig. 4). These figures illustrate the *ex-situ* and *operando* data sets are in excellent agreement at similar points in the charge cycle, showing very similar phase compositions (illustrated in Figure SI2 and highlighted in the main text lines 369-376), with minor exceptions as discussed in the SI. Based on these considerations, we are confident that the *ex-situ* data set is not strongly affected by charging phenomena.

To more clearly address these points Figure SI2 could be moved to the main text. For now to conserve space we have left Figure SI2 in the supporting information, however if the editor does not feel confined by space we will gladly promote it to the main text between Figure 4 and Figure 5.

Comment 5: I wonder if your results are affected by the SEI components having a different binding energy than the bulk of the sample? Should the SEI and the bulk peaks really be treated with the same calibration or is there a dipole effect in this interface? There are discussions about this in the literature based on HAXPES or tender XPS studies.

Response 5: The reviewer's comment is well taken, and in principle we do not rule out the possibility that interfacial charge-transfer effects might shift BEs of SEI phases. On the other hand, we did not see evidence for these interfacial effects in our experiments. The fundamental assumption made in the analysis of XPS spectra is that the Fermi level is uniform throughout samples when no *operando* bias (electron flood gun or UV light source) is applied. Since the binding-energy scale is defined in relation to the Fermi level ($E_F = 0$ eV), if this assumption holds then it should be valid to use the same BE calibration for both all layers, including LPS, SEI, and plated Li^0 layers. In fact, we observed that the Li 1s BE corresponding to Li^0 plated on the top-most surface is the same as that of sputter-cleaned pristine Li-foil samples discussed previously, i.e. $\text{BE}_{\text{Li}^0} = 55.0$ eV. This is what one would expect for a symmetric cell ($\text{Li}^0 / \text{SEI} / \text{LPS} / \text{SEI} / \text{Li}_{\text{foil}}$), where the net OCV across the structure should be zero. We take this as verification that the assumption of a uniform Fermi level under equilibrium conditions is valid. Nevertheless, the possibility raised by the reviewer that interfacial charging effects might affect measured SEI BEs is interesting and worth considering in future studies designed to look for these effects.

Please do not hesitate to contact me if additional information or materials are required. Thank you for your consideration of this article for publication.

Sincerely,

Glenn Teeter, Ph.D.
Materials Science Center
National Renewable Energy Laboratory
Colorado, United States of America

Reviewer #1 (Remarks to the Author):

The manuscript contains the new results and presents an interesting study on photoelectron spectroscopy. Since the paper is very well written, I recommend that this paper should be accepted after minor revision.

I suggest publishing the work if the following items are addressed:

Comment (1)

Figure 4 summarizes XPS spectra showing curve-fitting results. LPS and Li₂S fitting data peaks in the S 2p spectra indicate double peak. The double peaks were strange as fitting curve in the XPS analysis. Please address the correct compound in the manuscript.

In my opinion, the following revision is needed to make this manuscript suitable for publication.

For both the S 2p and P 2p core levels the 2p_{3/2} and 2p_{1/2} peaks have been combined into single lineshapes for simplicity. However, it should be noted that the appropriate intensity-ratio and BE separation constraints were applied during curve fitting of these features.

Comment (2)

Figure SI1 summarized EIS spectra showing the evolution of the SEI during the first three cycles of a Cu/LPS/Li coin cell. I think the figure which understands EIS data should be written clearly in order to measure initial conductivity. Please address this point.

The following conductivity provides very interesting information and it needs final revision to be acceptable for this Journal.

Fitting the component associated with the bulk of the electrolyte reveals the conductivity values for the LPS samples to be 1.61 mS/cm during first half cycle, 1.42 mS/cm after the second half cycle and 1.35 mS/cm after three full cycles. These results are summarized in the legend of Figure SI1